# Unveiling microbial succession dynamics on different plastic surfaces using WGCNA

**Keren Davidov**[ID]**, Sheli Itzahri, Liat Anabel Sinberger**[ID]**, Matan Oren**[ID]*

Department of Molecular Biology, Ariel University, Ariel, Israel

* matanor@ariel.ac.il

**Data Availability Statement:** All relevant data are within the manuscript and its Supporting information files.

**Funding:** This work was supported by the Israel Ministry of Science and Technology Israel-Portugal

## Abstract

Over recent decades, marine microorganisms have increasingly adapted to plastic debris, forming distinct plastic-attached microbial communities. Despite this, the colonization and succession processes on plastic surfaces in marine environments remain poorly understood. To address this knowledge gap, we conducted a microbiome succession experiment using four common plastic polymers (PE, PP, PS, and PET), as well as glass and wood, in a temperature-controlled seawater system over a 2- to 90-day period. We employed long-read 16S rRNA metabarcoding to profile the prokaryotic microbiome's taxonomic composition at five time points throughout the experiment. By applying Weighted Gene Co-expression Network Analysis (WGCNA) to our 16S metabarcoding data, we identified unique succession signatures for 77 bacterial genera and observed polymer-specific enrichment in 39 genera. Our findings also revealed that the most significant variations in microbiome composition across surfaces occurred during the initial succession stages, with potential intra-genus relationships that are linked to surface preferences. This research advances our understanding of microbial succession dynamics on marine plastic debris and introduces a robust statistical approach for identifying succession signatures of specific bacterial taxa.

## Introduction

Plastic pollution has become a global environmental concern, with far-reaching consequences for ecosystems, wildlife, and human health. Since its mass production began in the mid-20th century, plastic has transformed industries and daily life due to its durability, versatility, and cost-effectiveness. However, these very properties that make plastics invaluable have also led to their persistence in the environment. Each year, an estimated 8 to 12 million tons of plastic waste enters the oceans [1], where it accumulates, breaks down into microplastics, and infiltrates food webs [2, 3]. This accumulation not only threatens marine biodiversity but also carries potential risks to human health through bioaccumulation and trophic transfer [4].

Within minutes of entering the marine environment, plastic surfaces undergo colonization by marine bacteria [5]. Microbial colonization initiates a succession process marked by the formation of an initial biofilm that matures into a diverse community which includes phototrophs, heterotrophs, predators, decomposers, and symbionts [6]. This unique ecosystem, termed "plastisphere," supports biodiversity by offering a substrate for colonization in

collaboration Grant 3-1650 and the Israel Science Foundation (ISF) personal Grant 1556/23. the funders had no role in study design, data collection and analysis, decision to publish, or preparation of the manuscript.

**Competing interests:** The authors have declared that no competing interests exist.

nutrient-poor open waters and accordingly possesses a distinct taxonomic composition compared to the surrounding water [7]. The plastisphere allows certain microbial species to thrive, including those capable of metabolizing hydrocarbons and potentially degrading plastics. On the other hand, the plastisphere also enables the transport of invasive species and pathogenic microorganisms across vast distances, disrupting local ecosystems [8]. When plastisphere biofilms mature, they can influence the buoyancy of the floating plastic debris [9], affecting how they disperse and accumulate in the marine environment.

Understanding the temporal dynamics and processes driving microbial community assembly and succession is crucial for elucidating the structure, function, and interactions within these communities over time [10]. Investigating these processes within the plastisphere poses unique challenges due to the influence of variable environmental factors, including geography and season [11–14], immersion/incubation time [15, 16] and salinity [17]. Another variable to be considered is the composition of the plastic surface itself. Plastics are manufactured from diverse polymers, each possessing distinct densities and potentially containing various fillers and additives. Moreover, plastic debris within the ocean undergoes continuous weathering, influenced by light, UV radiation, mechanical forces, and biological factors [18]. Previous studies indicate that the state of plastic weathering significantly impacts the composition of the surface microbiome [19, 20]. Conversely, understanding the influence of plastic polymer types on microbiome development and composition has been challenging and often inconclusive [15, 21–23]. While some studies suggest polymer-related variations in the initial marine plastic microbiome [24–29], statistically robust colonization preferences and succession signatures of specific taxa within naturally formed biofilms are lacking.

In this study, we aimed to reveal the impact of polymer type on the colonization and succession dynamics of marine bacteria. We used pristine plastic pellets of four common plastic polymers: Polyethylene (PE), Polypropylene (PP), Polystyrene (PS), and Polyethylene Terephthalate (PET). Additionally, we included non-plastic beads (glass and wood) for reference. These pellets and beads were immersed in a controlled seawater system which was pre-inoculated with plastic-associated microbiomes from the Mediterranean Sea. Employing a straightforward experimental design, we utilized a MinION nanopore-based, long-read 16S rRNA metabarcoding pipeline, accompanied by a range of complementary data analyses, to explore trends in microbiome biodiversity and composition across the different surfaces. Weighted Gene Co-expression Network Analysis (WGCNA) was applied to the barcode sequence datasets to identify bacterial genera exhibiting significant surface- and time-specific succession signatures, as well as potential intra-genus relationships.

## Material and methods

### Experiments setup

The experiments took place in a 70-liter semi-sterile seawater aquarium with precise temperature control achieved through a chiller and a heater. The initial experiment, referred to as the 'primary' experiment, was conducted at 28 ± 2˚C, reflecting the average summer temperatures of the Israeli Mediterranean Sea coastal water (https://www.israelweather.co.il). Subsequently, a second experiment explored the temperature effect on the microbiome composition and was conducted at a higher temperature of 35 ± 2˚C, reflecting extreme conditions that may exist in shallow tidal pools during the summer [30]. Throughout both experiments, water salinity remained at approximately 40 parts per thousand (ppt), matching the typical salinity of the Eastern Mediterranean Sea (EMS) [31].

The aquarium was partitioned into two sections by a semipermeable barrier, facilitating water circulation while preventing the transfer of large items between compartments (S1 Fig

in S1 File). Before being filled with sterile artificial seawater, the aquarium, the barrier and additional parts were cleaned and sterilized with 6% sodium hypochlorite (bleach). Plastic-associated marine bacteria were inoculated into the aquarium by introducing plastic marine debris collected in Herzliya Marina, as detailed below. Six days after the introduction of the marine plastic debris, new surfaces were added to the left compartment of the aquarium, where water flowed back to the sump. These surfaces consisted of organza mesh bags made of polyester (MO45600, Apath international), 6 x 8 cm in size, with mesh holes size of approximately 0.2–0.4 mm. Each of the bags was filled with 7 grams of pellets/beads (3–5 mm in diameter) made of either pristine (pre-processed, pure) PE (Ipethene®, Carmel Olefines), PP (CAPILENE® W 77 AV, Carmel Olefines), PS (Styrolution® PS 124L, INEOS Styrolution), or PET (CZ-328, ZADE) plastic polymers, along with glass (5mm, 104017, Sigma) and wood (6 mm, LYS0118, Ningbo Linkyo). Before use, all materials were soaked in 70% ethanol for sterilization, followed by washing with sterile distilled-deionized water (DDW). The mesh bags, including replicates for each material according to the number of sampling time points, were suspended in mid-aquarium water by thin fishing lines tied to the aquarium top cover (S1 Fig in S1 File). In the second experiment, conducted at a higher temperature, we focused on PE, PET, glass, and wood to analyze the changes in the developing microbiome composition. All conditions, except the temperature, were similar to those of the primary experiment.

## Inoculation and sampling

To initiate the inoculation of plastic-associated marine bacteria, we collected various plastic debris items, including debris of cups, bags, straws, plastic cutlery, and bottles, from the water of Herzliya Marina (32˚ 09′ 38.8″ N 34˚ 47′ 35.0″ E). The debris was collected on 20 July 2021 using a sterile aquarium net. At the time of collection, the water salinity was 40 ppt, and the water temperature was ~29.2˚C. The debris was transferred with sterile tweezers into sterile glass vials containing filtered artificial seawater (FASW) that were kept in cooled Styrofoam boxes until reaching the lab. The collected debris underwent three washing steps with FASW to remove unattached material. After washing with FASW, the items were submerged in the aquarium compartment, receiving a continuous inflow of sump-filtered seawater (S1 Fig in S1 File).

For DNA extraction, pieces from four plastic debris items (D1-D4 in the primary experiment and D5-D8 in the second experiment) were sampled on the experiment onset. Sampling of the newly introduced pellets/beads took place at specific intervals: 2, 7, 14, 30, and 90 days from the beginning of the primary experiment and 7, 14, and 30 days from the start of the second experiment.

At each designated time point, one organza bag containing pellets/beads of each tested material and half a liter of the surrounding aquarium water were sampled. Before further processing, the pellets/beads underwent thorough washing in FASW, to remove un-attached bacteria. The water samples were filtered through a 0.22 μm polyethersulfone membrane using a 20 L/min pump (MRC). Subsequently, DNA extraction was conducted from both the pellets/beads and the membranes following the procedure described below. All procedures were conducted inside a laminar hood.

## DNA extraction

Pooled pellets/beads of each tested substrate, fragments of plastic debris, and membranes containing water filtrate underwent DNA extraction using the phenol-chloroform method. The extraction process was carried out in 15 ml tubes containing approximately 0.4 grams of sterile glass beads (G9268, Sigma) ranging in size from 425–600 μm, along with 3 ml of lysis buffer (composed of 10 mM Tris—HCl pH 8, 25 mM Na2EDTA pH 8, 1% v/v SDS, and 100 mM

NaCl). Samples underwent bead beating, followed by digestion with proteinase K (5 units/μL) and Lysozyme (2000 units/μL). Subsequent DNA extraction steps followed the protocol outlined in reference [32]. The eluted DNA was collected in 40 μL EB (10 mM TE Tris 1 mM EDTA pH 8). For wood samples, DNA extraction utilized the DNeasy PowerWater Kit (Qiagen) following the provided protocol to maximize DNA yield. All DNA samples were stored at -20˚C for subsequent analysis.

## Fourier-transform infrared spectroscopy (FTIR)

Pieces of the plastic items (D1-D8) retrieved from the environment underwent Fourier Transform Infrared (FTIR) analysis to determine their polymer composition (S2 Fig in S1 File). The FTIR analysis was conducted using a JACSOFT/IR-6800 spectrometer, with spectra collected in the wavelength range of 200 cm^-1 to 4000 cm^-1 and a fixed resolution of 2 cm^-1. Each sample's small plastic fragment was pressed under the K/Br crystal apparatus for measurement. The absorbance spectra were analyzed using the Open Specy open-source library [33].

## MinION library preparation and multiplexed nanopore sequencing

Sequencing runs were executed across three MinION flow cells in the primary experiment and one flow cell in the second experiment, accommodating 12–22 multiplexed libraries per run. Library preparation utilized the 16S barcoding kit (SQK-16S024, Oxford Nanopore Technologies) following the manufacturer's guidelines. For PCR amplification, 10 ng of genomic DNA from each sample was employed with barcoded primers targeting the full-length 16S rRNA gene (27F: 5′-AGAGTTTGATCMTGGCTCAG-3′ and 1492R: 5′-TACGGYTACCTTGTTA CGACTT-3′). All PCR procedures included PCR blanks to detect potential foreign DNA contamination in the water, reagents or primers. Subsequently, barcoded libraries were pooled, loaded onto MinION flow cells (FLO-MIN106D R9.4.1), and sequenced for 16–24 hours, with base-calling automated through the MinKnow program. Raw reads were acquired in both FAST5 and FASTQ formats, with only "pass" quality reads selected for subsequent analysis. All Nanopore MinION filtered reads analyzed in this project have been deposited in the NCBI SRA database under BioProjects PRJNA1005105 and PRJNA1012293.

## Data analysis

The base-called reads underwent processing using Nanopore version 2.1.0 and the 16S workflow v2023.04.21–1804452, accessed through https://epi2me.nanoporetech.com, via the EPI2ME desktop agent (ONT). The pipeline involved several steps: sorting reads into separate libraries based on multiplexing barcodes, trimming barcode and adaptor sequences, and comparison against the NCBI bacterial 16S database. Reads falling outside the range of 800 bp to 2000 bp in length were filtered out. Subsequently, the resulting CSV files from each EPI2ME run were downloaded and further processed using R (V4.3.1). Processing included merging EPI2ME 16s CSV outputs from individual MinION runs into a single file (pertaining to the primary experiment). Reads not assigned to a barcode and those lacking definitive classification (LCA≠0, either unclassified or belonging to more than one genus among the top three classifications) were filtered out. Multiplexing barcodes were converted into their corresponding samples, and read counts were normalized to relative abundances (percentages).

## Diversity analysis

Alpha diversity parameters were obtained using MicrobiomeAnalyst [34] based on taxonomic 16S metabarcoding datasets at the species level. Utilizing the mapped read counts of the

samples (S1 Table in S1 File) and observing species richness rarefaction curves (S3 Fig in S1 File), we standardized all samples to 7,940 mapped reads per sample. Community complexity within samples (alpha diversity) was evaluated using Chao and Shannon indexes to measure richness and diversity, respectively. Analyses were conducted based on sample type (combining all time points) and across time points (combining all surfaces) within the primary experiment. To assess dissimilarity in microbiome composition among samples (beta diversity), Principal Component Analysis (PCA) and Non-Metric Multidimensional Scaling (NMDS) with Bray-Curtis Dissimilarity Distances were employed. The PCA plot was generated using the PCAtools package (R package version 2.12.0, available at https://github.com/kevinblighe/PCAtools). An ANOVA test with post hoc pairwise comparison tests was performed to identify significant differences between the groups.

## WGCNA analysis

The WGCNA analysis was performed using the WGCNA R package v1.71 [35]. Hierarchical clustering was then performed on the samples, and samples with a cut height below 7000 were retained (S4A Fig in S1 File). Another filtering round was performed to remove bacteria with low variance (goodSamplesGenes (GSG) function within the WGCNA package). For the network topology analysis (TOM), soft-thresholding power $\beta = 10$ was chosen (S4B Fig in S1 File). First, a similarity matrix between each pair of bacteria across all samples was created. The similarity matrix was then transformed into an adjacency matrix, and the topological overlap matrix (TOM) along with the corresponding dissimilarity (1-TOM) value were calculated. Next, module eigengenes were calculated, and module-trait associations were quantified. Finally, the membership strength of individual species to their module and their significance to the relevant traits of that module was assessed. This allowed us to identify bacteria with relative preferences of surface type and time.

## Co-occurrence network visualization

The co-occurrence network analysis visualization was conducted using MicrobiomeAnalyst [34] with specific filter parameters: a minimum count of 20 and a mean abundance value of 30 (mbSet <- ApplyAbundanceFilter (mbSet, "mean", 20, 0.3)), and a low variance filter parameter of 10% (mbSet <- ApplyVarianceFilter(mbSet, "iqr", 0.1)). Only samples from the plastic microbiome were included in the analysis. The SparCC algorithm [36] was applied with a p-value threshold of 0.05 and a correlation threshold of 0.7 to identify significant co-occurrence patterns among microbial taxa.

## Results

### Metabarcoding outcomes and microbiome diversity

In total, 58 samples underwent sequencing, with 35 from the primary experiment, 15 from the second experiment, and eight from plastic debris (D1-D8). On average, each sample generated 55,378 reads (STDV = 32,465), of which an average of 36.4% were confidently mapped to the database in the primary experiment (with three top hits showing a single taxon, S1 Table in S1 File). The average read size across all samples was 1491.5 nucleotides (SDV = 23.5), indicative of the expected amplicon length for the nearly complete 16S rRNA gene sequence. When comparing alpha diversity parameters, plastic microbiome samples exhibited higher richness (p-value = 0.028) and biodiversity (p-value = 0.046) values compared to glass samples but lower than water samples (p-value = 1.931E-5 and 2.353E-5, respectively) (Fig 1A). No significant differences in alpha diversity parameters were observed among samples of different newly-

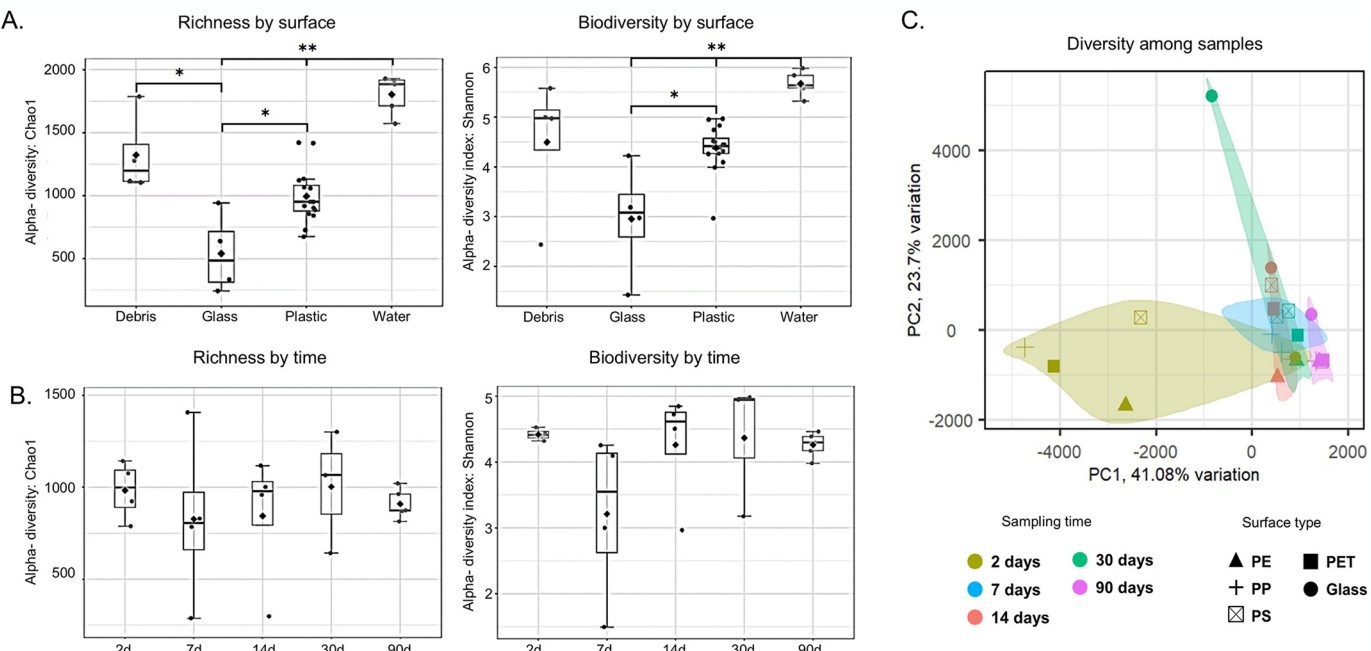

**Fig 1. Diversity parameters.** (A, B) Alpha diversity indexes for richness (Chao1) and biodiversity (Shannon). (C) PCA analysis for dissimilarity among samples (*ANOVA* test with post hoc pairwise comparison $^*p < 0.05$, $^{**}p < 0.01$).

introduced plastic polymers (S5 Fig in S1 File) or between them and the debris samples. Moreover, no significant differences were noted in alpha diversity indexes among different time points (all surfaces combined). However, the variance in Shannon index values was smaller at the two-day and 90-day time points compared to the 7, 14, and 30-day time points (Fig 1B, right).

Dissimilarities in the taxonomic composition of the microbiome among samples (beta diversity) were assessed through PCA analysis (Fig 1C). While the samples partially clustered according to time points, there was no clear clustering based on surface type. Principal component 1 (PC1, X-axis), explaining 41.08% of dissimilarities, highlighted the highest inter-sample diversity between the 2- and 90-day samples, with the 7, 14, and 30-day samples clustering together. Notably, the 2-day samples exhibited significantly higher dissimilarities among themselves compared to all other time points, indicating a more pronounced influence of surface type in the early stages of biofilm development.

## Plastic-attached microbiome becomes less specific with time

A diverse assortment of plastic items made of variable polymers (S2 Fig in S1 File) was introduced into the experiment to capture a wide range of plastic-associated bacteria for inoculating the new surfaces. This mixture comprised fragments or whole items like cups, bags, straws, plastic cutlery, bottles, and various other mostly fragmented plastic items.

To assess the migration of bacteria to the newly introduced surfaces, we compared the total number of mapped species from the sampled debris of the primary experiment (D1-D4) with that of all plastic surfaces at each time point. Results from the primary experiment revealed that 74.5% of the mapped species were shared between the debris and the plastic surfaces as early as two days after the experiment commenced, indicating rapid bacterial migration from

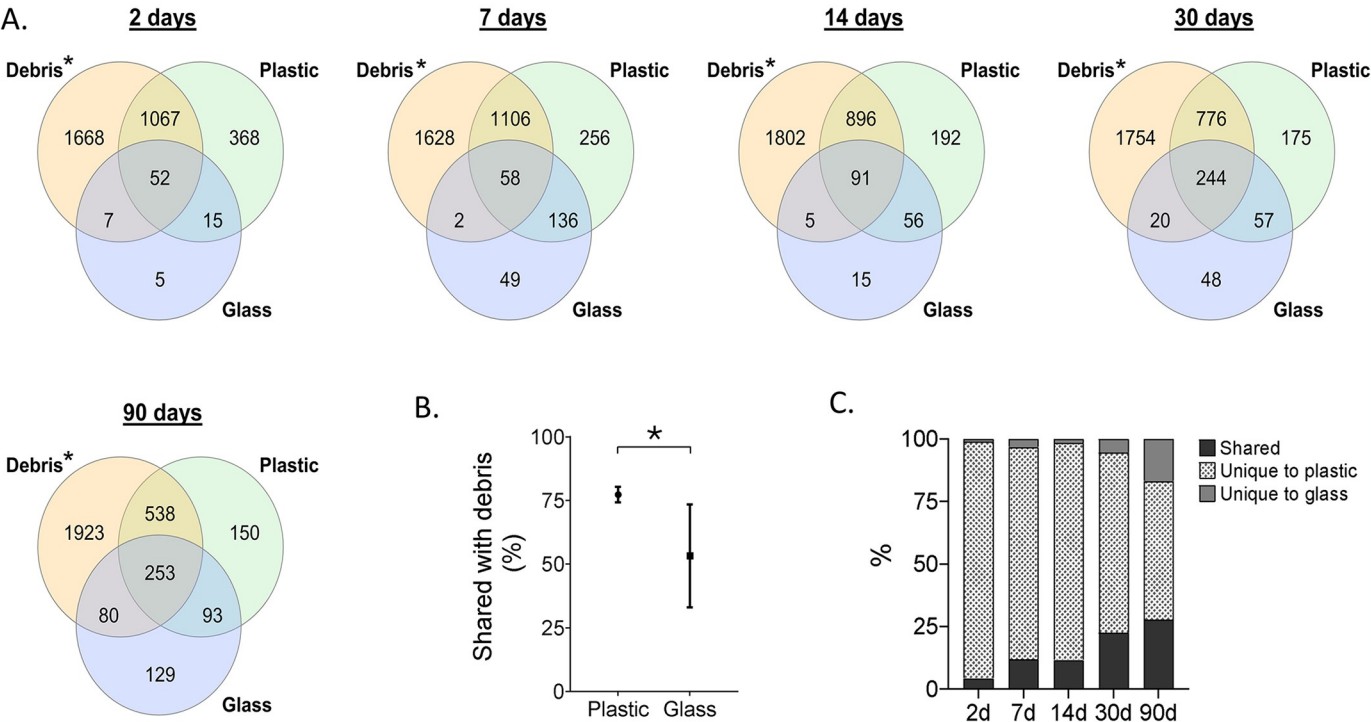

**Fig 2. Shared and unique mapped species between debris, plastic (all polymers combined), and glass.** (A) Venn diagrams by time points. (B) Percent of shared species between debris and plastic and between debris and glass (averages). (C) Percent of shared and unique species between plastic and glass at each time point. * Refer to the initial debris microbiome composition (at the beginning of the experiment).

the debris to the surfaces. This proportion remained consistent within the range of 68.4–78.1% throughout the experiment until 90 days (Fig 2A). It's noteworthy that these numbers may be underestimated as not all debris items were sampled.

Conversely, the proportions of shared mapped species between the glass surfaces and the debris were significantly lower (Fig 2B). Furthermore, comparison of mapped species lists between plastics and glass demonstrated a consistent trend of increasing proportions of shared species between the two surface types (Fig 2C), suggesting a gradual loss of microbiome surface-specificity.

## Bacterial relative abundance on different plastic polymers is similar at different temperatures

Identifying dominant taxa within a sample relies on their high read proportions relative to the total read count. In this study, we analyzed the top six most abundant genera within the pooled microbiome of each polymer, integrating data from all time points for that specific polymer (Fig 3). The average seawater temperature along the Mediterranean Israeli coast in the summer months (June-August) is ~28.5°C (https://www.israelweather.co.il/). However, the temperature in tidal pools, that often contain high concentrations of plastic debris, may reach as high as 40°C during the summer [30]. Accordingly, we performed the primary experiment at 28 ± 2°C and the second experiment in extreme 35 ± 2°C. Remarkably, despite the big temperature difference between the primary and the second experiments (7°C on average), nine out of sixteen identified dominant genera were shared between the two experiments and exhibited

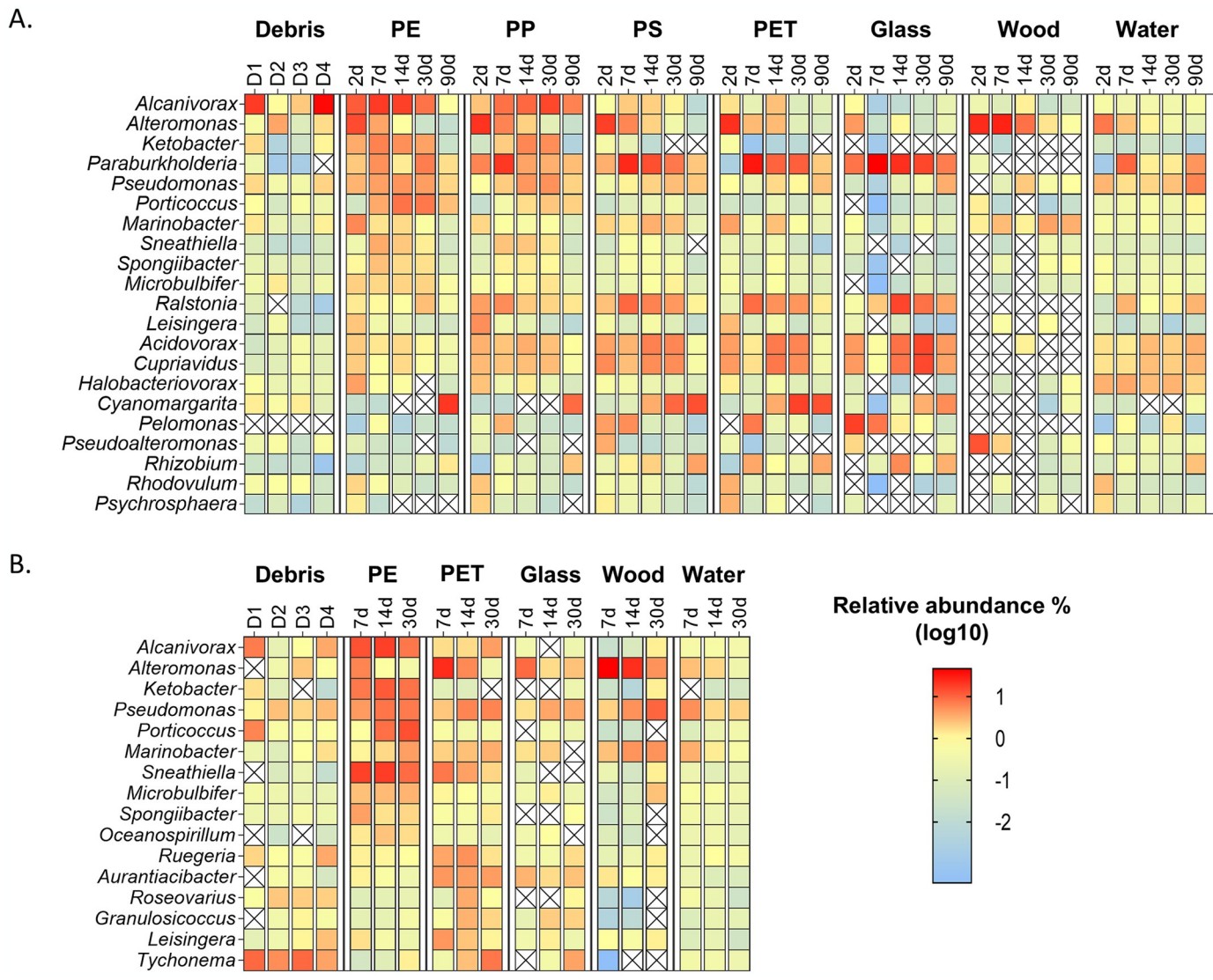

**Fig 3. Relative abundance of top abundant genera.** (A) Primary experiment at 28±2 ˚C. (B) Second experiment at 35±2 ˚C. The top 6 genera of each of the plastic polymer types are represented.

similar abundance dynamics in the two experiments. The shared genera included *Alcanivorax*, *Alteromonas*, *Ketobacter*, *Pseudomonas*, *Porticoccus*, *Marinobacter*, *Sneathiella*, *Microbulbifer* and *Spongiobacter*. In both experiments, *Alcanivorax*, *Ketobacter*, *Pseudomonas*, *Porticoccus*, *Sneathiella*, *Microbulbifer*, and *Spongiobacter* were more abundant in PE compared to PET, glass and wood between 7- and 30-days' time points and *Alteromonas* seems to be an early generalist colonizer in both experiments.

## Identification of pivotal genera within succession modules and genera with polymer-specific preference using WGCNA

We employed WGCNA to discern correlations among bacteria based on their relative abundances. This method facilitated the creation of a signed co-occurrence network for our

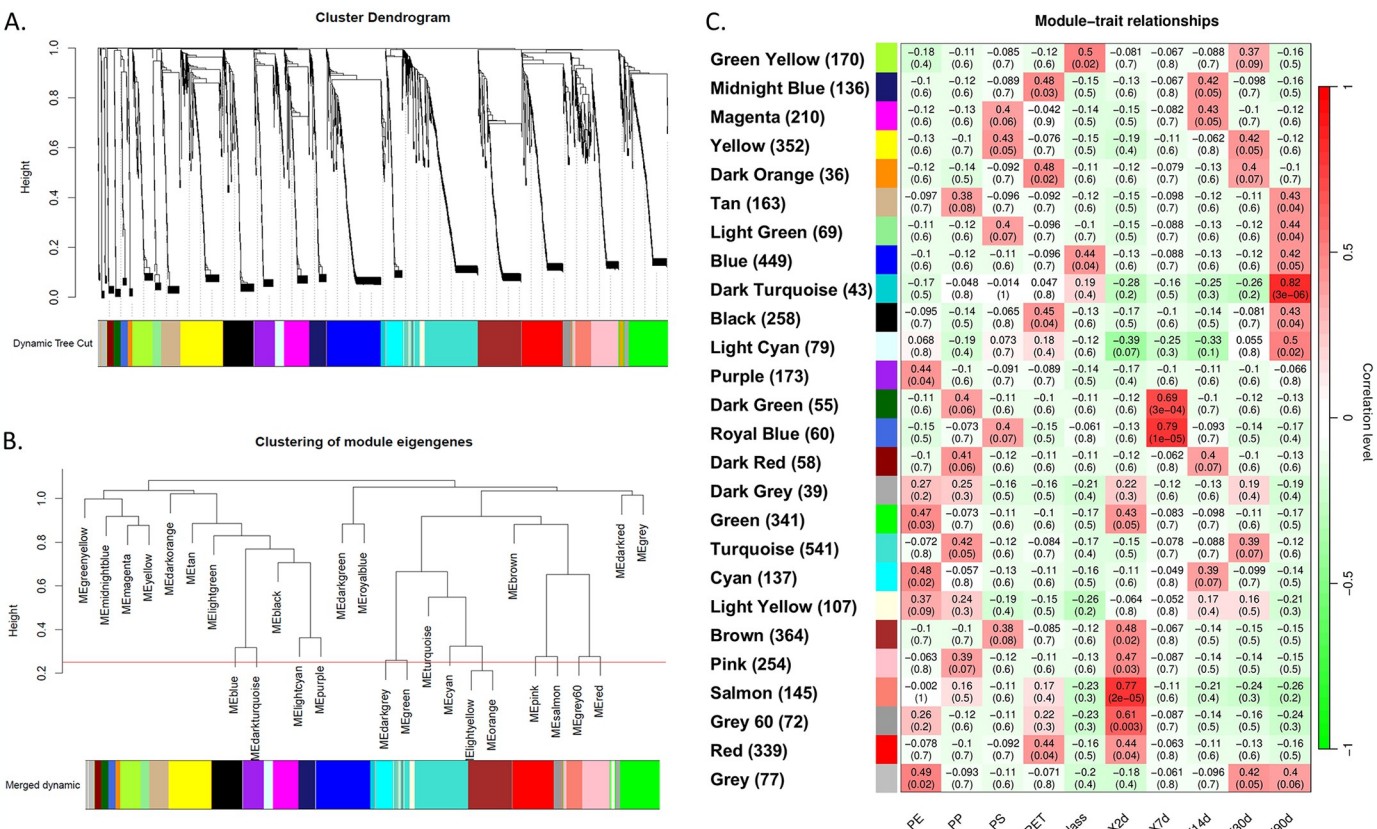

**Fig 4. Cluster dendrograms and module assignments.** (A) Cluster dendrogram based on the relative abundance patterns of identified bacterial species. Hierarchical clustering was performed using topology overlap measure (TOM). The colors represent the module assignments of the clusters. (B) Merged dynamic module clustering of similar eigengenes based on a merging threshold height = 0.25. (C) Module eigengenes table with module-trait corresponding correlation and significance values.

primary experiment. Hierarchical clustering of mapped bacterial species was then conducted, utilizing their read counts across various surfaces and time points (Fig 4A). Subsequently, module eigengenes were generated to capture clusters of bacterial species. To streamline the analysis, we performed a merged dynamic analysis (Fig 4B), resulting in 26 modules (indicated by distinct colors, Fig 4C). Module sizes ranged from 36 species (dark orange) to 541 species (Turquoise). A module-trait relationship table was constructed based on the module eigengenes. Notably, most module eigengenes exhibited positive correlations, often significant, to a single polymer and a specific time point. For instance, the Green Yellow module (Fig 4C, top row) displayed a positive correlation with glass and the 30-day time point (cor = 0.5, p-value = 0.02 and cor = 0.37, p-value = 0.09, respectively), while manifesting negative correlations with all other traits.

To identify pivotal genera within specific modules, we focused on species with a strong correlation to the module eigengenes (known as module membership) and associated traits (surface type or time). A genus was considered significant within a module if it comprised at least three species with a module membership score exceeding 0.9. For instance, *Alteromonas* emerged as a pivotal genus in the Salmon module, harboring 11 mapped species with a correlation exceeding 0.9 to the module (Fig 5A). Although this genus wasn't specifically associated with a particular surface, it displayed a significant correlation with the 2-day time point.

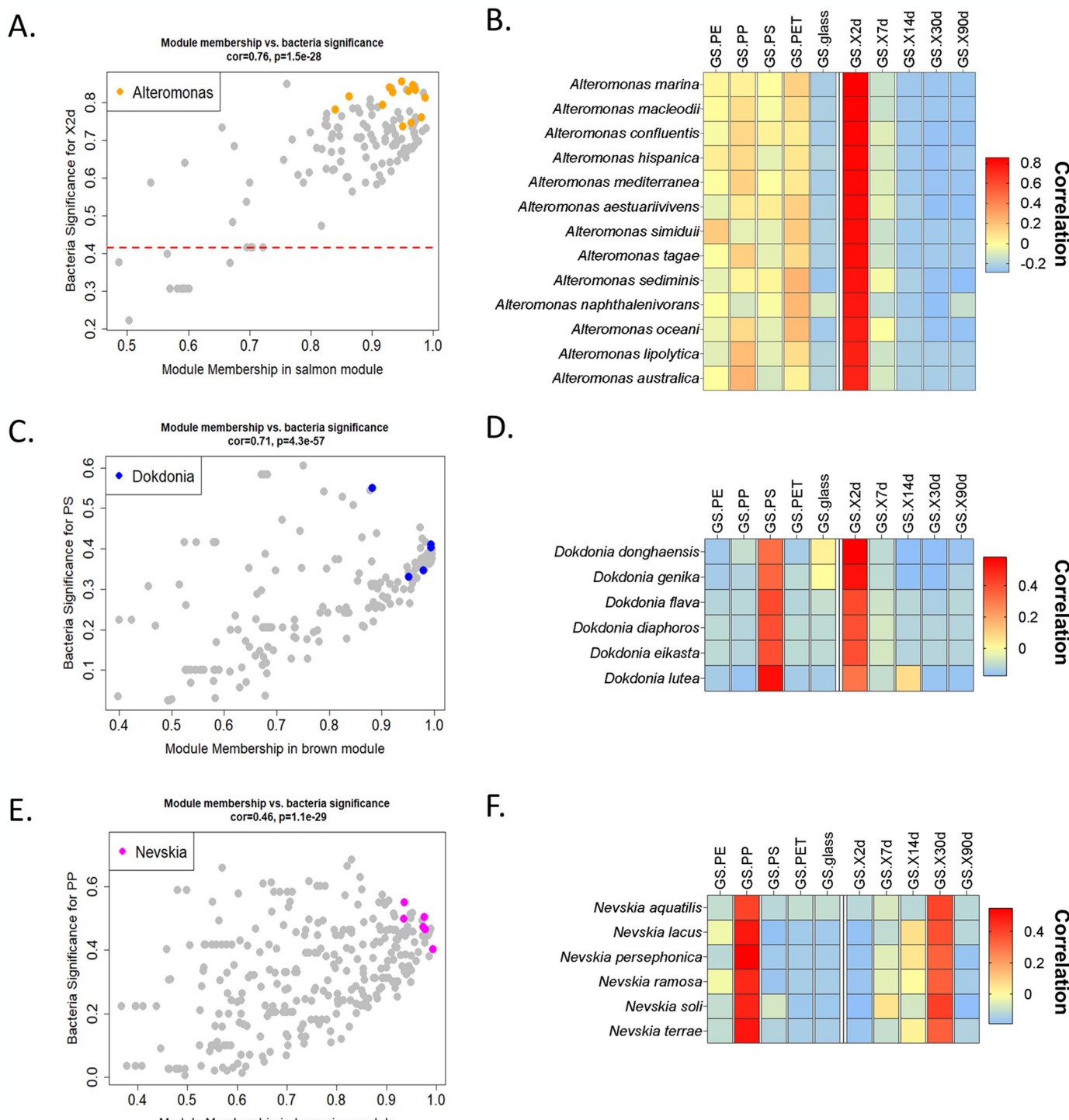

**Fig 5. Examples of pivotal genera within modules.** Module membership values of the key genera–*Alteromonas* (A), *Dokdonia* (B) and *Nevskia* (C) are represented in the context of their associate modules: Salmon, Brown and Turquoise accordingly (left side). The corresponding gene significance (GS) charts depicting the level of correlation of the key genera to all traits (surface type and time point) are presented on the right side. Dashed line indicates significance threshold (p-value = 0.05) to trait.

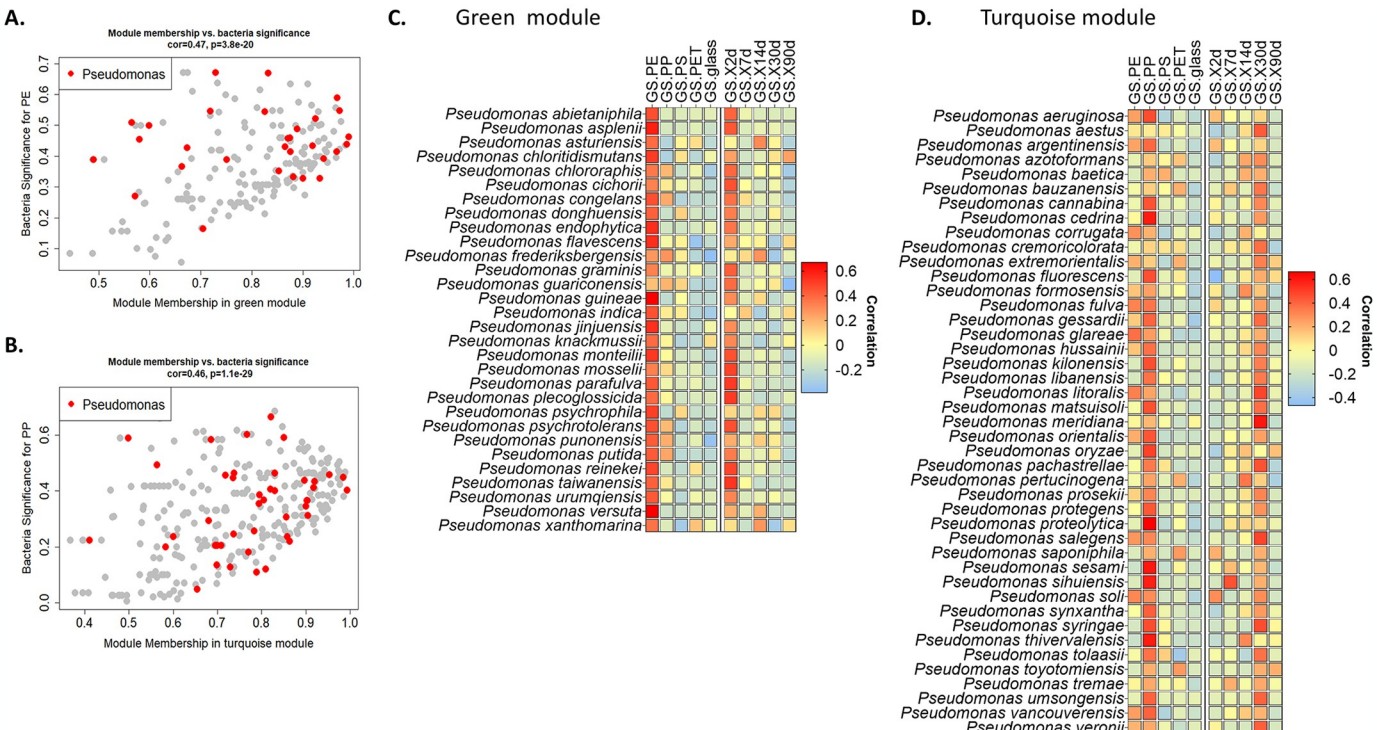

**Fig 6. *Pseudomonas* genus demonstrates succession sub-groups.** (A, B) Distribution of mapped species in modules Green (A) and Turquoise (B) by module membership and significance to the relevant trait (according to module eigengenes). *Pseudomonas* species are emphasized in Red. (C, D) The corresponding gene significance (GS) heat maps for *Pseudomonas*, depicting the level of correlation to each trait (surface types and time points).

Notably, most *Alteromonas* species within this module exhibited a negative correlation with glass and a positive correlation with all tested plastic surfaces (Fig 5A on the right), suggesting their role as early colonizers of various plastic polymers. Another example is the genus *Dokdonia*, which featured six mapped species within the brown module and demonstrated a robust positive correlation with the PS surface and the 2-day time point (Fig 5B), implying their early colonization on PS. Conversely, the genus *Nevskia*, encompassing six mapped species within the Turquoise module, exhibited a strong positive correlation with PP and the 30-day time point (Fig 5C), indicating a late colonization pattern on this polymer.

Several genera have been identified as pivotal across multiple modules. For example, the genus *Pseudomonas* exhibited pivotal roles in five distinct modules, notably prevalent in the green module (with 30 instances, 10 of which had module membership > 0.9, Fig 6A) and the turquoise module (comprising 43 mapped species, 9 with module membership > 0.9, Fig 6B). *Pseudomonas* in these two modules demonstrated a strong correlation with PE and the 2-day time point (Fig 6C), or with PP and the 30-day time point (Fig 6D), indicating the presence of multiple succession subgroups within the genus.

We identified 77 module-pivotal genera distributed across 21 modules (Table 1), with 17 modules demonstrating positive correlations with specific surface types and time points. Interestingly, glass-related pivotal genera within the green-yellow and blue modules were predominantly correlated with later succession stages (at 30-day and mainly 90-day time points), suggesting their absence during the early stages (Table 1).

To pinpoint genera enriched on specific plastic polymers, we conducted a screening process focusing on species demonstrating significant correlations with surface type traits compared

**Table 1. Pivotal genera by modules*.**

| Surface | Module | Genera | Time-related |
|---------|--------|--------|--------------|
| **PE** | Green | *Acinetobacter, Halomonas, Luteimonas, Marinobacter, Microbulbifer, Pseudomonas, Vibrio.* | 2d |
| | Cyan | *Marinomonas, Ochrobactrum, Pseudomonas.* | 14d |
| | Light yellow | *Pseudomonas.* | 14d, 30d |
| **PP** | Pink | *Aquimarina, Cystobacter, Methylobacterium, Nonlabens, Photobacterium, Vibrio.* | 2d |
| | Turquoise | *Acinetobacter, Aeromicrobium, Alcanivorax, Arenimonas, Bartonella, Desulfovibrio, Geobacter, Halomonas, Lysobacter, Nevskia, Nocardioides, Novosphingobium, Pseudomonas, Salinisphaera, Streptococcus, Streptomyces, Vibrio.* | 30d |
| | Tan | *Streptomyces.* | 90d |
| **PS** | Brown | *Dokdonia, Alteromonas, Aquimarina, Bradyrhizobium, Janthinobacterium, Kocuria, Methylobacterium, Methylorubrum, Microbacterium, Mycetocola, Photobacterium, Pseudoalteromonas, Staphylococcus, Thalassotalea, Vibrio, Winogradskyella.* | 2d |
| | Royal blue | *Burkholderia, Paraburkholderia.* | 7d |
| | Magenta | *Aquimarina, Flavobacterium, Streptomyces* | 14d |
| | Yellow | *Desulfovibrio, Rickettsia, Shinella, Streptomyces* | 30d |
| | Light green | *Streptomyces.* | 90d |
| **PET** | Red | *Celeribacter, Halomonas, Rhodovulum, Saccharospirillum, Shewanella, Simiduia, Vibrio, Winogradskyella.* | 2d |
| | Midnight blue | *Legionella, Staphylococcus.* | 14d |
| | Dark orange | *Paenibacillus.* | 30d |
| | Black | *Mesorhizobium, Paenibacillus, Pontibacter, Streptomyces.* | 90d |
| **Glass** | Green-yellow | *Acidovorax, Cupriavidus, Streptomyces.* | 30d |
| | Blue | *Achromobacter, Amylibacter, Aquabacterium, Chitinophaga, Clostridium, Dechloromonas, Exiguobacterium, Hydrogenophaga, Lewinella, Lysobacter, Massilia, Mesobacillus, Microbacterium, Planococcus, Propionispiram, Propionivibrio, Pseudomonas, Rhizobium, Shewanella, Sphingomonasm, Yoonia.* | 90d |
| **Surface-unspecific** | Salmon | *Alteromonas, Leisingera, Paraglaciecola, Phaeobacter, Psychrosphaera, Rhodovulum, Vibrio.* | 2d |
| | Grey 60 | *Marinobacter.* | 2d |
| | Light cyan | *Gloeocapsopsis.* | 90d |
| | Dark turquoise | *Achromobacter, Chitinophaga.* | 90d |

* Genera with at least three species with module membership score > 0.9.

to other surfaces within the initial month of the experiment (encompassing 2-day, 7-day, 14-day, and 30-day time points). We specifically targeted genera with a p-value ≤ 0.05 for surface significance (according to WGCNA results) and a threshold relative abundance exceeding 0.1%. Utilizing this filtration threshold, we identified 39 genera potentially enriched on specific plastic polymers, with 13 genera associated with PE, 11 with PP, 9 with PS, and 6 with PET (Table 2).

## The co-occurrence network shows the clustering of genera according to their polymer preferences

To deepen our comprehension of the relationships among various bacteria on plastic surfaces, we investigated the intra-genus connections within the co-occurrence network (Fig 7). The network was based on minimum threshold values for genus abundance, diversity among samples, and connection strength, as mentioned in the methods section.

The network visualization revealed four partially segregated clusters, with the major cluster labelled as "I" encompassing most of the genera, while the three smaller clusters (II, III, IV)

**Table 2. Significantly enriched genera on plastic polymers\*.**

| PE | PP | PS | PET |
|---|---|---|---|
| *Aestuariispira* (1) | *Actinomarinicola* (1) | *Granulosicoccus* (2) | *Catenovulum* (1) |
| *Alcanivorax* (2) | *Alcanivorax* (5) | *Kangiella* (1) | *Desulfomicrobium* (1) |
| *Amphritea* (1) | *Geobacter* (2) | *Marinobacter* (1) | *Kordiimonas* (1) |
| *Bdellovibrio* (1) | *Halioglobus* (2) | *Oceanibaculum* (1) | *Methyloversatilis* (1) |
| *Ketobacter* (1) | *Marinobacterium* (1) | *Paraburkholderia* (2) | *Pelagicoccus* (1) |
| *Marinicella* (1) | *Nevskia* (2) | *Pelomonas* (2) | *Ruegeria* (3) |
| *Marinobacterium* (2) | *Pseudomonas* (2) | *Pseudomonas* (1) | |
| *Microbulbifer* (3) | *Salinisphaera* (3) | *Rickettsia* (1) | |
| *Peredibacter* (1) | *Sandaracinus* (1) | *Thermogutta* (1) | |
| *Porticoccus* (1) | *Sneathiella* (1) | | |
| *Pseudomonas* (6) | *Tepidicaulis* (1) | | |
| *Spongiibacter* (1) | | | |
| *Umboniibacter* (1) | | | |

\* Genera with gene significant p-value $\leq 0.05$ and average relative abundance $\geq 0.1\%$ within a 30-days immersion period. The number of associated species is indicated in the parenthesis (the full list is in S3 Table in S1 File).

contained the remainder. Cluster I displayed a mix of genera with polymer-based abundance signatures, while clusters II, III, and IV appeared to harbour genera with more unified signatures. Cluster III featured seven genera significantly enriched on polyolefins (PE and PP), with *Alcanivorax* leading the cluster, followed by *Ketobacter* and *Sneathiella*. In contrast, Cluster II comprised four genera exhibiting relatively higher abundances on PS and PET, followed by PP, with *Paraburkholderia* emerging as the most prevalent genus within this cluster. Cluster IV housed four cyanobacteria genera with elevated abundance on PS and PET, followed by PE. The predominant genus within this cluster was *Cyanomargarita*.

## Discussion

Plastic debris found in the ocean comprises diverse combinations of polymers and additives, each endowed with unique chemical and physical properties. Despite their prevalence, little is known about how these properties influence the colonization and succession of microbial species. In this study, we investigated the successive development of bacterial microbiomes on four of the most abundant plastic polymer pollutants in marine environments: PE, PP, PS, and PET, along with two non-plastic abundant materials—glass and wood. The plastic polymers chosen exhibit distinct chemical structures; PE, PP, and PS feature a C-C backbone with varying side groups (hydrogen, methyl, and phenyl ring, respectively), whereas PET is a copolymer of ethylene glycol and terephthalic acid with an ester group in its backbone. These chemical disparities confer unique properties upon each polymer, including density, hydrophobicity, roughness, surface energy, and chemical reactivity, which collectively influence the adherence and growth of marine microorganisms.

It was shown that as the plastic biofilm cover develops, the effect of the polymer type on the microbiome composition is weakening until it is almost unnoticeable [6, 24]. This notion was validated in a recent study that showed that the polymer type affects microbiome composition and metabolic functionality in early marine plastic communities but not in mature communities [37]. Using linear regression analysis, we recently identified prokaryotic and eukaryotic taxa that were significantly enriched on specific plastic polymers in the marine environment [27].

Our findings show that greater variability is anticipated during the earlier developmental stages of plastic communities. Specifically, our results indicate that the various polymer

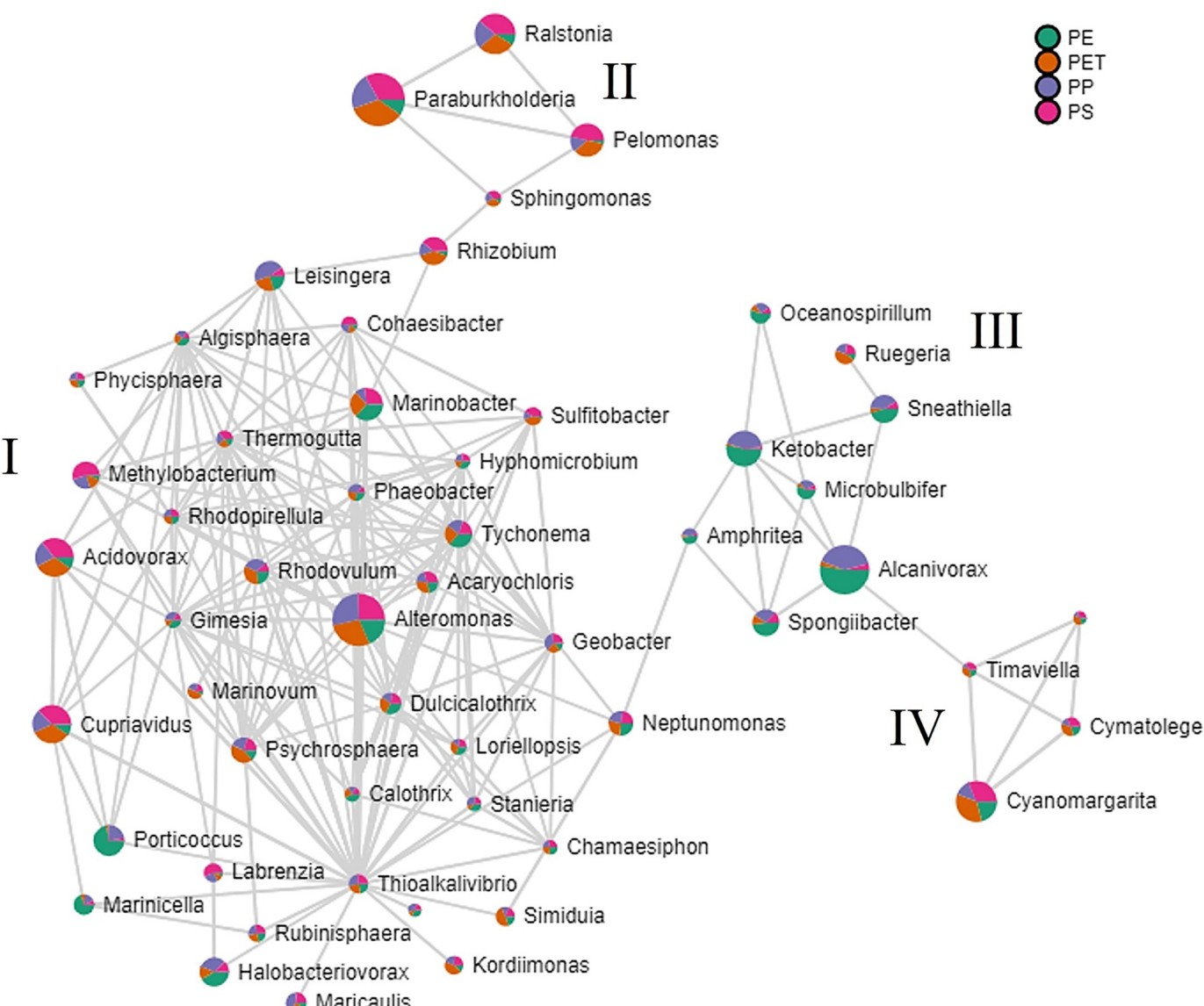

**Fig 7. Co-occurrence network of dominant plastisphere genera.** Only genera that passed the abundance filter are represented (see Methods section for details). The relative abundances of the represented genera on the different plastic polymers are shown in the pie charts in different colors: Green—PE, purple—PP, PET—orange and PS—pink. The pie chart sizes represent the general prevalence of each genus. Four identified clusters are marked with Roman letters (I-IV).

samples exhibited the highest dissimilarity at the 2-day time point, followed by a gradual, consistent shift toward equilibrium. This observation is further supported by the continual increase in the proportion of shared species between the plastic and glass surfaces over time.

To gain insights into the development of the plastisphere bacterial microbiome across different plastic and non-plastic surfaces, we initially screened our long-read 16S rRNA metabarcoding results for dominant genera based on their relative abundance on specific surface types and over time. While this analysis provided valuable data on highly abundant genera colonizing the surfaces, it lacked statistical significance validation. Upon comparing the lists of the prominent genera from the primary experiment with those from the second experiment, despite differences in temperature ranges (averaging 7°C), debris content from the

environment, and a smaller variety of polymers and the shorter time range in the second experiment, we observed that the majority of top-abundant genera (9 out of 16) in the second experiment were shared with those of the primary experiment. This suggests that polymer-based preference mechanisms are highly robust and less influenced by environmental factors.

Next, we utilized WGCNA to analyze the co-occurrence network of bacterial species within the microbiome of the different surfaces. WGCNA has been previously coopted for similar purposes, demonstrating its utility in deciphering complex interactions and associations within microbial communities [38–40].

The implementation of WGCNA in this study allowed us to delineate, for the first time, the succession signatures of plastisphere bacteria at both the genus and species levels, shedding light on their relationships with surface type and time-point dynamics. The validity of our WGCNA results was highlighted by the observation that multiple bacteria mapped to a specific genus were consistently clustered within specific modules, displaying high correlations with their respective module eigengenes in most instances. The findings of this study are reinforced by earlier research. For example, in concurrence with our results, previous studies have identified the genus *Alteromonas* as comprising pioneer generalist colonizer species [24, 41, 42]. Moreover, several bacterial genera, such as *Marinobacter* and *Marinobacterium*, which exhibited significant enrichment on plastic polymers in this study, are recognized as hydrocarbon metabolizers [43, 44]. Furthermore, at least three of the genera that were significantly enriched on PE, according to our analysis, were previously shown to degrade PE, including *Alcanivorax* [45], *Microbulbifer* [46] and *Pseudomonas* [47].

Using the MicrobiomeAnalyst platform, we identified four co-occurrence clusters of dominant bacteria, with three clusters containing genera exhibiting similar polymer-based occurrence signatures. The strong connections among genera within each cluster suggest tight collaborative interactions. We would like to note that while bacterial co-occurrence networks were achieved in this study, inter-kingdom taxa co-occurrence (i.e. between fungi and bacteria) are more challenging to decipher based on DNA metabarcoding analyses alone. This challenge is due to the differences in barcode regions for the different taxonomic groups. However, such interactions were previously reported within the plastisphere (i.e. [48]) and should be further explored.

We have shown in previous studies that using long, full-length 16S rRNA barcode sequences obtained by the nanopore MinION platform enables high-resolution taxonomic profiles of the plastisphere microbiome [49, 50]. The simple experimental design outlined in this study, coupled with the utilization of the nanopore-based metabarcoding pipeline and the analysis tools employed herein, holds practical applicability across various fields of microbiome research. For instance, it could be employed in the biomedical field to elucidate the colonization and succession patterns of pathogenic bacteria on artificial implants. Further fine-tuning of the analysis parameters will be necessary to address such future research inquiries. Nonetheless, we envision significant potential in the research approach presented here to gain deeper insights into microbial succession processes and preferences for specific surfaces or time frames.

## Conclusion

This study provides a comprehensive analysis of microbial colonization and succession on various plastic polymers in marine environments, revealing significant patterns in microbial diversity and community dynamics. Despite differences in polymer composition and environmental conditions, the findings highlight rapid bacterial migration to plastic surfaces, with over 74% of the microbial species shared between debris and newly introduced plastics within

just two days. This rapid colonization underscores the adaptability of marine microbial communities to this new surface type and their potential role in shaping the plastisphere.

The analysis of diversity parameters revealed that plastic surfaces support unique microbiomes with higher richness and biodiversity than non-plastic surfaces such as glass, but lower than surrounding water samples. Temporal trends indicated that early colonization stages exhibited the most pronounced microbial variability, gradually stabilizing over time. This observation, coupled with the identification of polymer-specific genera such as *Alcanivorax* and *Pseudomonas*, demonstrates the interplay between microbial communities and the chemical properties of plastic polymers.

By employing advanced network analysis methods such as WGCNA, this study identified pivotal bacterial genera and their succession dynamics, offering critical insights into the microbial interactions within the plastisphere. These findings contribute to our understanding of microbial ecology in plastic-polluted ecosystems and to developing targeted bioremediation strategies to mitigate the global plastic pollution crisis.

## Supporting information

**S1 File.**
(DOCX)

## Acknowledgments

The authors are grateful to Prof. Shiri Navon-Venezia for her scientific advice.

## Author Contributions

**Conceptualization:** Keren Davidov, Matan Oren.

**Data curation:** Keren Davidov, Sheli Itzahri, Liat Anabel Sinberger.

**Formal analysis:** Sheli Itzahri.

**Methodology:** Keren Davidov, Sheli Itzahri, Liat Anabel Sinberger, Matan Oren.

**Software:** Sheli Itzahri, Liat Anabel Sinberger.

**Supervision:** Matan Oren.

**Writing – original draft:** Keren Davidov, Sheli Itzahri, Matan Oren.

**Writing – review & editing:** Keren Davidov, Matan Oren.

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
