## [Decision Letter · Decision Letter 0]

25 Nov 2024

PONE-D-24-38682Unveiling microbial succession dynamics on different plastic surfaces using WGCNAPLOS ONE

Dear Dr. Oren,

Thank you for submitting your manuscript to PLOS ONE. After careful consideration, we feel that it has merit but does not fully meet PLOS ONE’s publication criteria as it currently stands. Therefore, we invite you to submit a revised version of the manuscript that addresses the points raised during the review process.

I have two reviews. The first one has sought  incorporation of  technical and methodological details. The first reviewer also provides several comments . I would like them to addressed by the authors in the revision of further consideration. The second review provides suggestions on how the manuscript can be be improvised and made useful to broader audience. The comments  made by the second reviewer are useful and suggest suitable consideration by the authors. 

We look forward to receiving your revised manuscript.

Kind regards,

Arga Chandrashekar Anil, Ph. D., D. Agr.,

Academic Editor

PLOS ONE

Journal requirements:    When submitting your revision, we need you to address these additional requirements. 1. Please ensure that your manuscript meets PLOS ONE's style requirements, including those for file naming. The PLOS ONE style templates can be found at https://journals.plos.org/plosone/s/file?id=wjVg/PLOSOne_formatting_sample_main_body.pdf and https://journals.plos.org/plosone/s/file?id=ba62/PLOSOne_formatting_sample_title_authors_affiliations.pdf 2. Please include a caption for table 1, 2 and figure 7.  3. In your Methods section, please provide additional information regarding the permits you obtained for the work. Please ensure you have included the full name of the authority that approved the field site access and, if no permits were required, a brief statement explaining why. 4. Thank you for stating the following financial disclosure:  [This work was supported by the Israel Ministry of Science and Technology Israel-Portugal collaboration Grant 3-1650 and the Israel Science Foundation (ISF) personal Grant 1556/23.].  Please state what role the funders took in the study.  If the funders had no role, please state: ""The funders had no role in study design, data collection and analysis, decision to publish, or preparation of the manuscript."" If this statement is not correct you must amend it as needed. Please include this amended Role of Funder statement in your cover letter; we will change the online submission form on your behalf. 5. Thank you for stating the following in the Acknowledgments Section of your manuscript: [The authors are grateful to Prof. Shiri Navon-Venezia for her scientific advice. This work was supported by the Israel Ministry of Science and Technology Israel-Portugal collaboration Grant 3-1650 and the Israel Science Foundation (ISF) personal Grant 1556/23.]We note that you have provided funding information that is not currently declared in your Funding Statement. However, funding information should not appear in the Acknowledgments section or other areas of your manuscript. We will only publish funding information present in the Funding Statement section of the online submission form. Please remove any funding-related text from the manuscript and let us know how you would like to update your Funding Statement. Currently, your Funding Statement reads as follows:  [This work was supported by the Israel Ministry of Science and Technology Israel-Portugal collaboration Grant 3-1650 and the Israel Science Foundation (ISF) personal Grant 1556/23.]. Please include your amended statements within your cover letter; we will change the online submission form on your behalf. 6. Please include captions for your Supporting Information files at the end of your manuscript, and update any in-text citations to match accordingly. Please see our Supporting Information guidelines for more information: http://journals.plos.org/plosone/s/supporting-information. 

Reviewers' comments:

Reviewer's Responses to Questions

**Comments to the Author**

1. Is the manuscript technically sound, and do the data support the conclusions?

Reviewer #1: Partly

Reviewer #2: Yes

2. Has the statistical analysis been performed appropriately and rigorously? 

Reviewer #1: Yes

Reviewer #2: Yes

3. Have the authors made all data underlying the findings in their manuscript fully available?

Reviewer #1: Yes

Reviewer #2: Yes

4. Is the manuscript presented in an intelligible fashion and written in standard English?

Reviewer #1: Yes

Reviewer #2: Yes

5. Review Comments to the Author

Reviewer #1: General comment

The subject matter is broadly interesting and although the subject matter (differences in microplastic biofilms) has been relatively well studied, there is some novelty in the use of longer whole 16S reads using nanopore, and the use of weighted correlation network analysis is also interesting. However, there are a number of technical details missing which make the paper as it is currently, unpublishable, and the missing experimental details need to be addressed before the paper can be accepted. The most egregious lacking information concerns how sterility of the experimental setup was achieved, and the handling of DNA extraction & PCR blanks. Without more information on these points specifically, the validity of the entire experiment is called into question. Generally, more justification for the statistical tests used and discussion concerning how different approaches might achieve different results is needed, especially concerning the use of WGCNA. Comparisons to other similar studies and their results needs to be conducted much more comprehensive, especially to other studies which have also used WGCNA.

Specific comments:

• Line 75 – spelling mistake – “litter”

• Why are these temperatures (28 and 35 degrees) chosen? Presumably these are water temperatures, although this should be stated, as well as how this was tracked throughout the experiment. Why so warm? What marine conditions are trying to be emulated? I am guessing conditions of the Herzliya coastal waters are trying to be emulated, since this is where the debris was apparently collected, but 35 degrees seems to be warmer that the water ever gets there (according to seatemperature.net). So why was this chosen? The experimental setup needs to be rationalised.

• Why use inoculated seawater as opposed to regular seawater? What were the conditions of the Herzliya marina at time of sampling? Similar water temperature and salinity to setup? What was the sampling date? How were the debris collected? How were they maintained between collection and inoculation to reduce microbial changes?

• How was all the equipment sterilized before use? Otherwise, what is the point of this inoculation approach if the organisms present in the setup are not of plastic origin?

• Were the plastic/wood/glass beads sterile before adding to the aquarium? If so how was this done. If not, how might the implicate the results?

• Why no PP or PS in the higher temperature situation? What are the sizes of the beads.

• Manufacturers of materials needs to be added in all cases, for setup equipment, plastics used, mesh bags etc.

• What chemical composition were the mesh bags? Were they plastic as well (organza can be a wide variety of materials, not scientifically accurate enough). What was the mesh size?

• Was post-sampling processing conducted in a sterile manner? There is no mention of a laminar flow cabinet being used. How was contamination managed?

• Why were wooden beads only subjected to a completely different extraction procedure. Different kits produce different results. Comparability can be problematic and certainly should be discussed.

• Line 155: How were sequences “compared to the NCBI database”. Using BLAST only? What date/what version of the database

• No mention of extraction or PCR blanks in methods. This is highly problematic for publication and must be addressed before the paper can be published.

• Lines 231-236 this information should really be in methods

• Line 256 – why is the temperature delta 7 degrees on average? I thought this was accurately controlled and kept consistent (although it is not explained how this was maintained and monitored, which again is problematic)

• Why are the modules named after colours? It reads as slightly ridiculous. Why can’t they have alpha/numeric identifiers?

• Line 320 (and Table 2 generally): Relative abundance greater than just 0.1% doesn’t sound like enrichment. Why such a low cutoff?

• Table 2: Regarding the number in brackets, number of associated species generally? i.e. 6 of the total number of Pseudomonas species were specific to PE? Or is 6 species all the Pseudomonas species found in the study? It would be important to show average relative abundance of these taxa to show the reader if they really are enriched or simply being detected by chance here and not on other surfaces. What is the P-value based off? What statistics was done? Indicator analysis? If so explain, if not, what analysis was used and why?

• Why use long read as opposed to something shorter but covering more sequence depth, like V3-V4 with Illumina, which is more commonly used. This difference, including what can be better seen with long reads vs shorter barcoding methods should be discussed! It is one of the main differentiating factor compared to other works and should be discussed in depth

• Line 388-390 – Similarly, since WGCNA is the also a major thing that is new/interesting about the study, discussing previous uses of this and how this newer approach enables new insights is also very important to be discussed. Needs to be expanded significantly

• I would expect a conclusion section, but this is absent

Reviewer #2: Thank you for sharing your manuscript. I found the study valuable in advancing our understanding of microbial succession on different plastic polymers in marine environments. The identification of polymer-specific microbial communities and the successional trends across environments are particularly compelling. However, I believe the manuscript would benefit from some adjustments and additions that explain the significance in greater depth.

In particular, your introduction and discussion sections are quite short and lack depth. Contextualizing the broader implications of your findings and expanding on how they contribute to addressing the global plastic pollution crisis would be beneficial. Below are some suggestions for improvement:

Broader Implications:

- Global Context: While the results are well-documented, the manuscript does not sufficiently situate these findings within the larger context of global plastic pollution. It would strengthen the study to articulate how understanding microbial succession on plastics could influence degradation predictions or provide insight into the long-term ecological impacts of persistent plastic waste.

- Ecosystem Effects: The potential for microbial colonization to alter nutrient cycling, carbon fluxes, or interactions with higher trophic levels is an area that could be explored further. For example, could the dominance of certain genera, such as Alcanivorax, signal broader ecological consequences, such as shifts in hydrocarbon degradation pathways?

- Environmental and Policy Relevance: Emphasizing the environmental consistency of polymer-specific communities is valuable, but further discussion on its significance would be helpful. Could these findings inform efforts to design more sustainable plastics or predict ecological risks across environments?

- Plastic Degradation Potential: The identification of genera like Pseudomonas and their potential to influence biodegradation pathways presents an opportunity to discuss the dual implications of microbial activity: both as a possible avenue for mitigating plastic pollution and as a vector for environmental risks (e.g., transport of pathogens or invasive species).

Deepening the Interpretation of Results:

- Ecological Roles of Genera: While the study identifies key microbial genera, there is limited discussion on their ecological functions or roles in shaping succession patterns. For instance, why do some genera dominate later stages of colonization? Are they exploiting specific metabolic niches or competitive advantages within the biofilm matrix?

- Successional Patterns: The observed patterns are well-described, but their drivers are less clear. Additional insights into the mechanisms driving succession (e.g., nutrient availability, surface properties of plastics) would enhance the interpretation.

Tying Findings to Broader Significance:

I recommend linking your results explicitly to the broader ecological and societal implications in the discussion and conclusion. For example:

- In the introduction, briefly outline how understanding microbial colonization can contribute to mitigating the impacts of plastic pollution or developing sustainable solutions.

- In the discussion, dedicate a section to synthesizing the ecological significance of these findings and their relevance to global efforts to address plastic pollution.

- In the conclusion, consider a forward-looking statement about how this research could inform future studies or practical applications, such as plastic design or environmental management.

Lastly there are some small edits I suggest you make e.g explainign why you chose the temperatures you used in your incubation experiments. Furthermore, the use of glass and wood as comparative surfaces in your experiment is a sound choice, as these materials are often used as controls in Plastisphere studies. However, to ensure that the experimental design is clearly understood, I recommend explicitly stating that these materials are being used as controls. This would help to clarify the intent behind including them and would better contextualize the significance of the microbial communities found on plastic surfaces in comparison to those on other materials.

By elaborating on these points, your manuscript could move from being a strong descriptive study to one with a larger impact on how we understand and address plastic pollution in marine environments.

6. PLOS authors have the option to publish the peer review history of their article (what does this mean?). If published, this will include your full peer review and any attached files.

Reviewer #1: No

Reviewer #2: No

---

## [Author Response · Author response to Decision Letter 0]

15 Dec 2024

Dear Dr. Arga Chandrashekar Anil, 

Academic Editor, PLOS ONE.

We want to express our gratitude to you and the reviewers for your thorough evaluation and constructive feedback on our manuscript entitled "Unveiling microbial succession dynamics on different plastic surfaces using WGCNA" The insightful comments and suggestions provided have been invaluable in enhancing the quality and clarity of our work. In this document, we have addressed each reviewer's comments in detail . Our responses include explanations of the revisions, justifications for the approaches taken, and additional data or references where necessary. We hope our revisions meet the reviewers' expectations and further strengthen the manuscript.

General comments

Response: The MS was re-checked and adjusted to meet PLOS ONE's style requirements.

2. Please include a caption for tables 1, 2 and figure 7. 

Response: Captions for tables 1 and 2 added. The caption for Figure 7 was present. 

Response: No permit was required. We have positioned the samples in places that did not interfere with the normal operation of the Marina.

 [This work was supported by the Israel Ministry of Science and Technology Israel-Portugal collaboration Grant 3-1650 and the Israel Science Foundation (ISF) personal Grant 1556/23.]. 

Response: The statement is correct.

[The authors are grateful to Prof. Shiri Navon-Venezia for her scientific advice. This work was supported by the Israel Ministry of Science and Technology Israel-Portugal collaboration Grant 3-1650 and the Israel Science Foundation (ISF) personal Grant 1556/23.]

Please remove any funding-related text from the manuscript Response: removed and let us know how you would like to update your Funding Statement. Currently, your Funding Statement reads as follows: 

 [This work was supported by the Israel Ministry of Science and Technology Israel-Portugal collaboration Grant 3-1650 and the Israel Science Foundation (ISF) personal Grant 1556/23.].

Response: The statement is correct.

Reviewer #1: General comment

The subject matter is broadly interesting and although the subject matter (differences in microplastic biofilms) has been relatively well studied, there is some novelty in the use of longer whole 16S reads using nanopore, and the use of weighted correlation network analysis is also interesting. However, there are a number of technical details missing which make the paper as it is currently, unpublishable, and the missing experimental details need to be addressed before the paper can be accepted. The most egregious lacking information concerns how sterility of the experimental setup was achieved, and the handling of DNA extraction & PCR blanks. Response: We accepted Reviewer #1's comment and added details about how sterility was kept throughout the MS. With that being said, we would like to mention that the whole setup was built on land with sterile artificial seawater that was made from dry salt, so the chances for contamination of marine bacteria seem very low. Nevertheless, we took extra measures to ensure sterility, as detailed in the revised version. Without more information on these points specifically, the validity of the entire experiment is called into question. Generally, more justification for the statistical tests used and discussion concerning how different approaches might achieve different results is needed, especially concerning the use of WGCNA. Comparisons to other similar studies and their results needs to be conducted much more comprehensive, especially to other studies which have also used WGCNA. Response: We accept this comment and refer to it in the revised version.

Specific comments:

1. Line 75 – spelling mistake – “litter” . 

Response: Corrected.

2. Why are these temperatures (28 and 35 degrees) chosen? Presumably these are water temperatures, although this should be stated, as well as how this was tracked throughout the experiment. Why so warm? What marine conditions are trying to be emulated? I am guessing conditions of the Herzliya coastal waters are trying to be emulated since this is where the debris was apparently collected, but 35 degrees seems to be warmer than the water ever gets there (according to seatemperature.net). So why was this chosen? The experimental setup needs to be rationalised.

Response: While the Eastern Mediterranean Sea (EMS) seawater temperatures rich ~30°C in August, the temperature in tidal pools along the coasts rich up to 40°C and even more (reference added). Since many of these pools contain high concentrations of plastic debris, we expect that plastic-attached bacteria will be exposed to those conditions as well. A section was added (Lines 80-89).

3. Why use inoculated seawater as opposed to regular seawater? Response: Using artificial seawater allowed us to maintain sterility, adjust salinity parameters and make sure there are no contaminations that may temporarily exist in natural seawater. What were the conditions of the Herzliya marina at time of sampling? Response: Details were added (Line 116-117). Similar water temperature and salinity to setup? Response: Yes. What was the sampling date? Response: July 20th 2021 (added to text, Line 116). How were the debris collected? Response: Using a sterile aquarium net. How were they maintained between collection and inoculation to reduce microbial changes? Response: In cooled Styrofoam boxes until reaching the lab (Added to text, Line 119)

4. How was all the equipment sterilized before use? Otherwise, what is the point of this inoculation approach if the organisms present in the setup are not of plastic origin? Response: All equipment, including the aquarium itself, was cleaned and sterilized either with bleach or ethanol. During the experiment all aquarium parts were covered to prevent dust or liquid droplets from entering. Because the setup was based on artificial sterilized seawater and it was separated from any marine contamination, we are assured that no contamination could influence the results. Moreover, our DNA metabarcoding results showed hits of strictly marine or dual (such as E. coli) bacteria and no terrestrial bacteria. 

5. Were the plastic/wood/glass beads sterile before adding to the aquarium? If so how was this done? If not, how might they implicate the results?

Response: Yes we added a sentences (Line 104-105): “All materials were soaked in 70% ethanol for sterilization, followed by washing with sterile distilled-deionized water (DDW).”

6. Why no PP or PS in the higher temperature situation? What are the sizes of the beads.

Response: Due to financial constraints associated with nanopore sequencing, PP and PS samples were not included in the second experiment. From these three, PE is the most abundant floating plastic polymer and it was chosen for this experiment. Bead/pellets sizes ranged from 3–5 mm in diameter for plastic beads, 5mm for glass and 6mm for wood beads, as described in the Methods section (Line 100-104).

7. Manufacturers of materials need to be added in all cases, for setup equipment, plastics used, mesh bags, etc.

Response: Manufacturer details have been added to the methods section for all materials and equipment used (Line 100-104).

8. What chemical composition were the mesh bags? Were they plastic as well (organza can be a wide variety of materials, not scientifically accurate enough). What was the mesh size?

Response: The mesh bags were made of polyester with a mesh size of approximately 0.2–0.4 mm. This information has been clarified in the methods section (Line 98-100).

9. Was post-sampling processing conducted in a sterile manner? There is no mention of a laminar flow cabinet being used. How was contamination managed?

Response: Post-sampling processes were conducted under sterile conditions in a laminar flow cabinet. Line 135:”All procedures were conducted inside a laminar hood” Plastic debris and mesh bags were washed with sterile artificial seawater (ASW) to remove unattached materials. Sterile equipment was used throughout.

10. Why were wooden beads only subjected to a completely different extraction procedure. Different kits produce different results. Comparability can be problematic and certainly should be discussed.

Response: DNA extraction from wood using the phenol-chloroform method resulted in low yield and poor quality. Therefore, a commercial DNA extraction kit yielded higher concentration and better purification. Due to these differences, wood samples were excluded from most analyses and are only presented in Figure 3.

11. Line 155: How were sequences “compared to the NCBI database”. Using BLAST only? What date/what version of the database

Response: The 16S rRNA sequences were compared to the NCBI bacterial 16S database using the EPI2ME 16S workflow provided by Oxford Nanopore Technologies. This workflow utilizes a proprietary algorithm for taxonomic assignment. The specific version of the workflow used was v2023.04.21-1804452, accessed via the EPI2ME desktop agent. The database used by the workflow corresponds to the NCBI bacterial 16S database.

12. No mention of extraction or PCR blanks in methods. This is highly problematic for publication and must be addressed before the paper can be published.

Response: PCR blanks are conducted in all our PCR routines to test for primer and foreign DNA contaminations. We don’t think this obvious control should be mentioned in the MS.

13. Lines 231-236 this information should really be in methods

Response: Accepted. This information has been moved to the methods section (line 204-216)

14. Line 256 – why is the temperature delta 7 degrees on average? I thought this was accurately controlled and kept consistent (although it is not explained how this was maintained and monitored, which again is problematic)

Response: This line refers to Figure 3, which illustrates the temperature differences between the primary experiment (28°C) and the secondary experiment (35°C) as mentioned in the text. We made modifications to clarify it in the text. 

15. Why are the modules named after colours? It reads as slightly ridiculous. Why can’t they have alpha/numeric identifiers?

Response: WGCNA (Weighted Gene Co-expression Network Analysis) assigns modules color names as a default convention to facilitate easy distinction. For reference, see Langfelder and Horvath (2008): DOI: 10.1186/1471-2105-9-559.

16. Line 320 (and Table 2 generally): Relative abundance greater than just 0.1% doesn’t sound like enrichment. Why such a low cutoff?

Response: The cutoff of 0.1% was just the threshold cutoff of the data that was used for the WGCNA analysis to prevent errors related to random contaminations. We added the word “threshold” to clarify this. 

17. Table 2: Regarding the number in brackets, number of associated species generally? i.e. 6 of the total number of Pseudomonas species were specific to PE? Or is 6 species all the Pseudomonas species found in the study? It would be important to show average relative abundance of these taxa to show the reader if they really are enriched or simply being detected by chance here and not on other surfaces. What is the P-value based off? What statistics was done? Indicator analysis? If so explain, if not, what analysis was used and why?

Response: The numbers in brackets indicate the species specifically associated with each polymer type based on WGCNA results. For example, 6 Pseudomonas species (out of ~180 in the study) were associated with PE. Table 2 and Supplementary Table S3 were generated using WGCNA results, combined with relative abundance data, 

We have updated Supplementary S3 Table to include relative abundance values for clarity. The P-values are derived from WGCNA correlations. 

18. Why use long read as opposed to something shorter but covering more sequence depth, like V3-V4 with Illumina, which is more commonly used. This difference, including what can be better seen with long reads vs shorter barcoding methods should be discussed! It is one of the main differentiating factor compared to other works and should be discussed in depth

Response: We have added a secession in the discussion mentioning the advantage of nanopore full 16s sequencing (Lines 484-489)

19. Line 388-390 – Similarly, since WGCNA is the also a major thing that is new/interesting about the study, discussing previous uses of this and how this newer approach enables new insights is also very important to be discussed. Needs to be expanded significantly

Response: The advantages and novelty about the use of the technique in this study is described in the discussion (Lines 456-459)

20. I would expect a conclusion section, but this is absent

A conclusion section was added.

Reviewer #2: 

Thank you for sharing your manuscript. I found the study valuable in advancing our understanding of microbial succession on different plastic polymers in marine environments. The identification of polymer-specific microbial communities and the successional trends across environments are particularly compelling. However, I believe the manuscript would benefit from some adjustments and additions that explain the significance in greater depth.

In particular, your introduction and discussion sections are quite short and lack depth. Contextualizing the broader implications of your findings and expanding on how they contribute to addressing the global plastic pollution crisis would be beneficial. Below are some suggestions for improvement:

Response: We thank the reviewer for this comment and expended our introduction and discussion sections accordingly.

Broader Implications:

- Global Context: While the results are well-documented, the manuscript does not sufficiently situate these findings within the larger context of global plastic pollution. It would strengthen the study to articulate how understanding microbial succession on plastics could influence degradation predictions or provide insight into the long-term ecological impacts of persistent plastic waste.

- Ecosystem Effects: The potential for microbial colonization to alter nutrient cycling, carbon fluxes, or interactions with higher trophic levels is an area that could be explored further. For example, could the dominance of certain genera, such as Alcanivorax, signal broader ecological consequences, such as shifts in hydrocarbon degradation pathways?

- Environmental and Policy Relevance: Emphasizing the environmental consistency of polymer-specific communities is valuable, but further discussion on its significance would be helpful. Could these findings inform efforts to design more sustainable plastics or predict ecological risks across environments?

- Plastic Degradation Potential: The identification of ge

---

## [Editor Report · Decision Letter 1]

19 Jan 2025

PONE-D-24-38682R1Unveiling microbial succession dynamics on different plastic surfaces using WGCNAPLOS ONE

Dear Dr. Oren,

Thank you for submitting your manuscript to PLOS ONE. After careful consideration, we feel that it has merit but does not fully meet PLOS ONE’s publication criteria as it currently stands. Therefore, we invite you to submit a revised version of the manuscript that addresses the points raised during the review process.

**I have one minor point to be made regarding point 12 of the comments offered by reviewer 1: the authors can  include a statement to manuscript and not simply brush aside the obvious.**

We look forward to receiving your revised manuscript.

Kind regards,

Arga Chandrashekar Anil, Ph. D., D. Agr.,

Academic Editor

PLOS ONE

**Journal Requirements:**

**Additional Editor Comments:**

I have one minor point to be made regarding point 12 of the comments offered by reviewer 1: the authors can include a statement to manuscript and not simply brush aside the obvious.

---

## [Author Response · Author response to Decision Letter 1]

21 Jan 2025

Dear Dr. Arga Chandrashekar Anil, 

Academic Editor, PLOS ONE.

We thank you for providing us with the opportunity to revise our manuscript, entitled "Unveiling microbial succession dynamics on different plastic surfaces using WGCNA" We are grateful for the feedback and have carefully addressed all points raised during this second round of review. Below, we provide detailed responses to the comments and outline the corresponding revisions made to the manuscript.

Editor Comments:

1. I have one minor point to be made regarding point 12 of the comments offered by reviewer 1: the authors can include a statement to manuscript and not simply brush aside the obvious.

Response: This comment refers to reviewer 1 comments: "No mention of extraction or PCR blanks in methods. This is highly problematic for publication and must be addressed before the paper can be published." 

In our initial response, we explained that PCR blanks are routinely used in all our PCR workflows to test for primer and foreign DNA contamination and indicated that we did not think it was necessary to explicitly state this in the manuscript. However, we now understand the importance of providing this information explicitly to ensure transparency and to align with the journal’s expectations. Accordingly, we have revised the Methods section to explicitly mention the inclusion of PCR blanks. The updated text now states: “All PCR procedures include PCR blanks to detect potential foreign DNA contamination in the water, reagents or primers" (line 166). We believe this addition addresses the concern and provides the necessary detail to ensure clarity for readers.

Journal Requirements:

Response: We have thoroughly reviewed the reference list to ensure that all citations are accurate and complete.

We trust that these revisions and responses address all concerns raised by the reviewers and the editorial team. We appreciate your continued guidance throughout this process and are happy to address any further questions or requirements.

For the Authors,

Keren Davidov & Matan Oren

---

## [Editor Report · Decision Letter 2]

23 Jan 2025

Unveiling microbial succession dynamics on different plastic surfaces using WGCNA

PONE-D-24-38682R2

Dear Dr. Oren,

We’re pleased to inform you that your manuscript has been judged scientifically suitable for publication and will be formally accepted for publication once it meets all outstanding technical requirements.

Kind regards,

Arga Chandrashekar Anil, Ph. D., D. Agr.,

Academic Editor

PLOS ONE
---

## [Editor Report · Acceptance letter]

28 Jan 2025

PONE-D-24-38682R2 

PLOS ONE

Dear Dr. Oren, 

I'm pleased to inform you that your manuscript has been deemed suitable for publication in PLOS ONE. Congratulations! Your manuscript is now being handed over to our production team.

Kind regards, 

on behalf of

Professor Arga Chandrashekar Anil 

Academic Editor

PLOS ONE